# Oral Complications from Oropharyngeal Cancer Therapy

**DOI:** 10.3390/cancers15184548

**Published:** 2023-09-14

**Authors:** Vidya Sankar, Yuanming Xu

**Affiliations:** Department of Diagnostic Sciences, Tufts University School of Dental Medicine, Boston, MA 02111, USA; yuanming.xu@tufts.edu

**Keywords:** oropharyngeal cancer, dysgeusia, xerostomia, radiation caries, osteoradionecrosis, salivary gland dysfunction, mucositis

## Abstract

**Simple Summary:**

Although oropharyngeal cancers are increasing in incidence, more patients are surviving their cancer therapy. Once a patient has completed their oropharyngeal cancer treatment, they need to contend with the multiple side effects related to their cancer treatment. These factors may be acute or chronic and vary from individual to individual, depending on the cancer stage, location, and treatment modalities employed. These treatment-related side effects have an impact on their overall survival and quality of life domains such as function, taste, speech, dry mouth, dental decay, oral infection, bone necrosis, and nutrition. This review will summarize the most common oral complications from oropharyngeal cancer therapy, their causes, ways to help reduce the occurrences, and guidelines related to monitoring and treatment of the conditions.

**Abstract:**

Post-oropharyngeal cancer treatment complications include a multitude of oral side effects that impact overall survival and quality of life. These include acute and chronic conditions affecting the oral cavity and head and neck, such as mucositis, infection, xerostomia, dysgeusia, radiation caries, osteonecrosis, and trismus. This review will summarize the most common oral complications from oropharyngeal cancer therapy. The authors would like to point out that the literature cited frequently combines oropharyngeal and head and neck cancer results. If recommendations are made strictly related to oropharyngeal cancers, this will be highlighted.

## 1. Introduction

Oropharyngeal cancers are increasing in incidence; it is estimated that new cases and deaths will be 54,540 and 11,580, respectively, in 2023 [1]. There were 16.9 million cancer survivors in the United States in 2019, with projected increases to 22.2 million by 2030 [1]. Although the number of cases is increasing, the risk of death from oropharyngeal cancer is decreasing, broadly attributed to overall decreases in smoking, early detection, surgical techniques, and the implementation of targeted therapies [2].

These gains in survival come with acute and long-term consequences associated with side effects from their cancer treatment, including organ damage and functional impairments, which vary depending on the particular cancer treatment modality, the intensity of treatment, and/or combinations of treatment modalities. Methods used to reduce treatment-related side effects include the use of minimally invasive surgical techniques when possible. These include transoral robotic surgery (TORS) and transoral laser microsurgery (TLM). Compared with open surgery, a smaller incision is made, and telescopes and surgical blades are inserted. The approach allows for greater precision in tumor resection, leading to decreased post-operative pain, shorter hospital stays, less scarring, decreased impact on speaking and swallowing, a reduction in trismus, and, in some cases, a reduction in chemotherapy and/or radiation therapy (RT). Other methods employed to reduce treatment side effects include intensity-modulated radiation therapy (IMRT) and proton beam therapy. These modalities allow for more effective delivery of radiation therapy to the area of interest (the tumor). IMRT uses computer-controlled linear accelerators to deliver radiation to the three-dimensional shape of the tumor or field of interest with high precision, sparing/minimizing damage to surrounding anatomical structures. Proton beam therapy is a form of external beam RT using particle accelerators that create a focused beam of protons instead of photons. The protons produce less scatter by stopping photon beams immediately (Bragg peak effect) after they deliver their peak energy to the treatment site, with a rapid decline in energy release beyond this depth. This results in less impact on surrounding tissue [3]. Other targeted cancer treatment approaches include novel therapeutics that inhibit tumor growth and/or metastasis. For head and neck cancers, these include epidermal growth factor receptor (EGFR) inhibitors and immune checkpoint inhibitors targeting programmed death 1 (PD-1) and programmed death ligand 1 (PD-L1). Despite the use of these methods, oral complications occur.

This article will present the most common oral complications in survivors of oropharyngeal cancer therapy. These include acute adverse events (mucositis, candida infection, salivary gland dysfunction) and related sequela (taste alterations, radiation caries), which may be acute or chronic. Additionally, late-term adverse events like osteoradionecrosis (ORN) and trismus will also be reviewed. We will also discuss the rationale and timing related to cancer surveillance and factors clinicians of the head and neck should consider when planning patient scheduling and follow-up. 

## 2. Oral Mucositis

Oral mucositis is a common and distressing complication frequently experienced by oropharyngeal cancer patients undergoing chemotherapy and/or RT. This condition involves inflammation of the oral mucosa, typically manifesting as erosions and ulcers within the oral cavity. Mucositis causes patients severe pain and discomfort, necessitating the use of narcotic analgesics, a reduction in oral intake, and dysphagia with subsequent nutritional inadequacy, prolongation of hospitalization, and a significant increase in financial burden [4].

The incidence of oral mucositis resulting from RT or chemoradiation in patients with head and neck cancer varies based on the treatment regimen employed, patient-specific characteristics, and the site of the cancer being targeted. Overall, the incidence ranges from 59% to 100%, with severe oral mucositis ranging from 23% to 81% [4,5]. While proton therapy has been associated with a reduction in severe oral mucositis by minimizing integral doses, the incidence of oral mucositis still ranges from 11% to 58% [6,7].

Clinically, oral mucositis starts as focal erythema with sensitivity of the oral mucosa, progressing to ulceration with irregular borders, pseudomembranous covering, and exudate accompanied by severe pain. Traditional cytotoxic chemotherapeutic agents, such as cisplatin and 5-FU, tend to affect non-keratinized areas, such as the buccal mucosa, lip mucosa, soft palate, lateral and ventral tongue, and floor of the mouth, with a temporal relation to the initiation of chemotherapy. Mucositis peaks around 2 weeks after infusion and will typically heal, then recur with each chemotherapy cycle. Radiation-related mucositis affects the area within the radiated field, with signs beginning at an accumulated dose of 15 Gy (around 10 days), progressing to full severity at 30 Gy, and is maintained until the completion of the RT [8]. The majority of radiation-induced oral mucositis tends to heal within 3 weeks after the completion of treatment, but chronic oral mucosal lesions, defined as lesions lasting for more than 3 months, have been reported. Features of chronic oral mucositis include erythema, atrophy, ulceration, and telangiectasias, with a prevalence of 8% [9,10].

The clinical presentation and course of mucositis can be complicated by the combination of a variety of chemotherapeutic agents with radiation. For example, epidermal growth factor receptor (EGFR) monoclonal antibodies (cetuximab) and immune checkpoint inhibitors (ICI), such as pembrolizumab or nivolumab, produce variable manifestations, including lichenoid mucositis (mixed white striations, erythema, and/or ulcerations), vesiculobullous features, erythema multiforme, or Stevens–Johnson syndrome/toxic epidermal necrolysis (SJS/TEN) [11,12,13]. The onset of immunotherapy-associated oral adverse events can range from weeks to months after the first initiation [14].

The pathogenesis of mucositis is multifactorial, involving the interaction of various cellular and molecular events triggered by cancer treatments. A recent review summarized individual characteristics associated with the severity of oral mucositis induced by chemotherapy or radiation, including age, gender, oral hygiene status, salivary gland function, physiological parameters, and genetic characteristics [15]. A comprehensive exploration of the molecular pathogenesis of oral mucositis is beyond the scope of the current paper, but several recent reviews have elucidated advancements in our understanding of this topic [16,17].

Though the mechanism could be complicated, a simplified model with key molecular changes has been well recognized to explain the pathophysiology of chemotherapy- or RT-induced oral mucositis [18]. These five stages, summarized in the Sonis article, are listed in Table 1.

Various classifications of mucositis severity exist, which rely on clinical manifestations and symptoms. Some also include the ability to function and/or the necessity for adjunctive interventions (Table 2). The most common scoring system in clinical trials is the World Health Organization (WHO) scoring system, valued for its straightforwardness, direct applicability, and extensive adoption within both clinical contexts and research endeavors. 

Contemporary strategies for managing oral mucositis predominantly revolve around prevention and the amelioration of symptoms. A pivotal element in the prevention of oral mucositis lies in the meticulous design of RT protocols with a focus on safeguarding non-affected oral mucosal surfaces. The employment of RT techniques that fastidiously preserve the integrity of healthy tissues can notably curtail both the frequency and intensity of oral mucositis. Additional strategies for oral mucositis management are outlined in evidence-based guidelines developed by the Mucositis Study Group of the Multinational Association of Supportive Care in Cancer and the International Society of Oral Oncology (MASCC/ISOO) [19,20]. 

The MASCC/ISOO guidelines recommend that basic oral care should be applied to head and neck cancer patients undergoing chemotherapy or radiation therapy for oral mucositis prevention. Basic oral care includes professional dental evaluation and treatment before cancer therapy to reduce the risk of odontogenic infections. Saline and sodium bicarbonate rinses can enhance oral clearance of debris, improve oral hygiene, and enhance patient comfort. Anti-inflammatory agents, such as benzydamine mouthwash, are strongly recommended to prevent oral mucositis during moderate-dose radiation therapy (<50 Gy) for head and neck cancer patients. This recommendation extends to patients receiving chemoradiation therapy, albeit with weaker evidence. Intraoral photobiomodulation therapy with low-level laser therapy is also recommended for oral mucositis prevention, particularly in radiation therapy patients.

Furthermore, alternative strategies to prevent oral mucositis in head and neck cancer patients who have undergone RT or chemoradiation therapy include the use of honey and oral glutamine. Caution is warranted when employing oral glutamine due to the observed higher mortality rates in patients who received parenteral glutamine supplementation. Regarding the management of pain associated with oral mucositis (OM) in head and neck cancer patients after chemoradiation therapy, a 0.2% topical morphine mouthwash is suggested [19,20]. 

Contemporary investigations and clinical trials involving small molecular agents are prominently directed toward targeting elements of the innate immune response and pivotal molecules implicated in the initiation and propagation of inflammation. These studies particularly highlight targets such as superoxide dismutase mimetics, Nrf2 activators, and NF-kB modulators [21]. These findings point towards a promising trajectory in the field of oral mucositis management.

## 3. Candidiasis 

Oropharyngeal candidiasis is prevalent among head and neck cancer patients, particularly those with oropharyngeal malignancies. Risk factors include hyposalivation, altered oral microbial flora, compromised immune function, and denture-wearing [22]. Candidiasis in the oral and oropharyngeal regions can present as white removable papules with erythematous bases (pseudo-membrane candidiasis), focal erythematous patches (erythematous candidiasis), fissures and erythema on the corner of the mouth (angular cheilitis), and hyperplastic white plaques on the oral mucosa (hyperplastic candidiasis) [23,24,25,26,27]. *Candida albicans* is the predominant species, while other fungal species such as *C. tropicalis*, *C. glabrata*, and *C. dubliniensis* also occur but are less prevalent [26,28]. 

A diagnosis based solely on Candida colonization cultures may not accurately indicate active candida infection, given the opportunistic and commensal nature of candida species [29]. Therefore, diagnosis should rely on clinical signs and symptoms. Empirical treatment can be used for patients with oral manifestations and symptoms. Topical treatment, including topical polyenes (nystatin and amphotericin) and azoles (clotrimazole), proves effective as a first-line approach for immune-competent individuals. For immune-compromised patients or those with severe infections, systemic antifungal medications such as fluconazole and voriconazole should be considered. Culture and antifungal drug sensitivity testing are needed for patients with chronic or refractory candidiasis. Antifungal prophylaxis should be considered for patients with a heightened risk of oropharyngeal candidiasis, particularly those who are immunocompromised, have a history of recurrent infections, or are undergoing treatments that further compromise their immune system’s functionality. Loo et al. reviewed antifungal agents in oral candidiasis prevention and identified clotrimazole as the most effective agent when compared to placebo, with fluconazole being the safest [30]. A recent systematic review comparing fluconazole with other antifungal agents showed no significant difference in the outcome [31]. Table 3 presents a summary of common antifungal regimens, aiding in clinical decision-making for optimal patient care [32,33].

## 4. Salivary Gland Dysfunction 

Salivary gland dysfunction (SGD) is one of the most common complications of head and neck cancer treatment, both from chemotherapy (acute) [34] and RT (acute and chronic). The role of saliva is well documented. The mucins found in saliva serve several important functions within the oral cavity. These functions include (1) making it less susceptible to abrasive trauma; (2) flushing the oral cavity to reduce debris and microorganisms; (3) dissolving tastants (see Taste Dysfunction section); (4) containing digestive enzymes; (5) neutralizing acids; (6) providing protection of the dentition (see Caries section); (7) containing proteins and peptides with antibacterial, antiviral, and antifungal effects; and (8) providing factors that enhance wound healing [35]. The parotid and submandibular glands contribute equally to about 90% of total daily saliva production, while the minor salivary glands supply the remaining amount. Newly detected tubarial glands were discovered in 2020; however, their contribution to total salivary flow rates is unknown [36]. The submandibular/sublingual glands primarily contribute to the production of resting saliva, whereas mastication stimulates flow from the parotids predominantly. Salivary flow rates differ according to gender, medical co-morbidities, medications used to treat systemic diseases, dehydration, and circadian rhythms.

Treatments for oropharyngeal cancers include a combination of surgery and chemotherapy, RT, and/or targeted therapy. While all of these therapeutic modalities target cancer cells, they also cause damage to healthy surrounding tissue to varying degrees. Surgical procedures that impact oral anatomical structures, including salivary glands, muscles, and nerves, contribute to salivary gland impairment, neurologic deficiencies, and other functional impairments. These effects result in difficulty forming food boluses, speaking, swallowing, and may result in esthetic concerns. These side effects have both acute and chronic consequences. 

RT uses high-energy particles to kill or arrest cancer cells along with other tissue within the field of treatment, including salivary glands, adjacent nerves, vascular structures and endothelium, and bone, leading to acute and long-term consequences including salivary gland dysfunction, mucositis, oral infections, ORN, and taste dysfunction. Salivary gland reduction contributes to a remarkably high caries rate known as radiation caries. Salivary gland tissue is extremely radiosensitive. When damaged, it results in irreversible impairment, but it is unclear if this is due to the direct effects of radiation on the acini and ducts or due to injury of the surrounding vascular structures, fibrosis, and/or inflammation. Parotid glands exhibit changes at radiation doses of 24–26 Gy and up to 39 Gy for submandibular glands [37]. Irradiated submandibular glands result in decreased acinar area, angulation, vacuoles, increased spaces between acini in parotid glands, and ductal degeneration [38]. Salivary hypofunction occurs in over 80% of head and neck cancer patients [39]. Salivary gland function decreases initially from 1.3 mL/min to approximately 0.2 mL/min during the first 1–3 months post-RT. It rebounds to 0.4 mL/min at 3–6 months, and this level is maintained for >2 years post-RT. It never returns to pre-treatment levels [40]. Traditional measures to minimize damage to surrounding tissues include modulation of fractionated RT, IMRT, volumetric modulated arch therapy, proton therapy, the use of intraoral stents, and salivary gland relocation [41]. 

New experimental methods to minimize damage under investigation include pre-treatment with cell cycle inhibitors, a reduction in reactive oxygen species generation, dysregulated calcium signaling, controlling inflammatory responses, inhibition of autophagy/reduction in apoptosis, treatments that increase endothelial cell division and capillary content, gland regeneration through salivary stem/progenitor cells, and insertion of aquaporin water channels [41,42]. According to a recent review of salivary gland hypofunction and/or xerostomia induced by nonsurgical cancer therapies by MASCC/ISOO and the American Society of Clinical Oncology (ASCO), some treatment modalities have strong evidence, while with others, there is limited or insufficient evidence. A summary of the MASCC/ISOO guidelines is presented in Table 4 [43].

Treatments for xerostomia and diminished salivary flow include maintenance of adequate hydration, parasympathomimetics (pilocarpine and cevimeline), topical coating agents (artificial saliva, lubricating gels, sprays, and mouthwashes), masticatory/gustatory stimulants (sugar-free gums, candies, lozenges, and mucoadhesive agents), room humidifiers, and avoidance of mouth breathing. Patients with thick salivary secretions may benefit from mucolytic agents such as guaifenesin. 

## 5. Taste Dysfunction 

Taste dysfunction after cancer therapy is common and reported to affect up to 93% of patients receiving chemotherapy [44,45], rebounding to 47% and 48% at 6 months and 1 year after treatment, respectively [39]. Additionally, 100% of those receiving RT to the head and neck experience taste dysfunction and the degree [46] is proportional to the RT dose delivered [47]. Taste dysfunction with RT also rebounds over time. In RT patients, taste dysfunction begins the first week of treatment, peaking at the third to fourth week of treatment; partial improvement begins several weeks after the end of RT and returns to near normal within one year post-treatment [48].

Taste dysfunction contributes to weight loss and a reduced quality of life. A recent review by Galaniha and Nolden [49] looked at the role of salivary dysfunction and its effects on taste dysfunction in cancer patients. Briefly, saliva production is stimulated by circadian patterns (peaking at 3–5 p.m. and lowest at 4 a.m.) [50], masticatory, gustatory, and olfaction stimulation. Saliva solubilizes tastants from food and delivers them to taste buds mostly through the fungiform papilla of the tongue. Additionally, through the mechanical forces of chewing and movement of the tongue, tastants are distributed to the circumvallate and foliate papilla and palatal mucosa. The more hydrophobic the food, the less dissolution there is in saliva and the diminished taste sensation. Not only does saliva deliver tastants, but the mucins in saliva also protect the taste receptors from damage through friction and microorganisms. It was found that diminished salivary flow rates and removal of the major salivary glands resulted in shrinkage of tastebuds, suggesting saliva plays a role in taste transduction and the maintenance of taste papillae [51,52].

Non-pharmacologic liposomal agents, polysaccharide-based oral rinses, lactoferrin supplementation (chemotherapy), ginger, parasympathomimetics (bethanechol, pilocarpine, cevimeline), zinc supplements, and artificial salivas have been shown to improve taste dysfunction [49]. 

## 6. Radiation Caries

Patients who have undergone RT for head and neck cancer encounter an elevated susceptibility to dental caries. The maintenance of tooth health is intricately governed by salivary functions, which encompass pH regulation, remineralization promotion, and the provision of antimicrobial and cleansing effects. The RT-induced impairment of salivary gland function, leading to hyposalivation, markedly exacerbates the vulnerability of head and neck cancer patients to dental caries [53]. Additionally, radiation therapy exerts a cascading impact on dental structures [54,55]. It precipitates enamel and dentin degradation, triggers shifts in oral microflora with concomitant pH alterations, and influences patient diet and oral hygiene practices, collectively escalating the risk of rapid and rampant caries [56,57]. Beyond the intrinsic factors, treatment-associated variables also play a pivotal role in enhancing caries susceptibility. Factors such as the dosages of radiation administered and the history of concurrent chemotherapy are known contributors to the heightened vulnerability of caries within this patient cohort [58].

Moore et al. found that 37% of patients developed RC within the first two years of their cancer therapy, which was radiation dose-dependent [58]. Brennan et al. conducted a prospective study that followed 572 head and neck cancer patients over 24 months, of whom 45.8% were diagnosed with oropharyngeal cancer. They found that the percent change in decayed, missing, and filled surfaces (DMFS) increased at 0, 6, 12, 18, and 24 months post-RT was 47.6% (1.3), 48.7% (1.3), 50.1% (1.3), 51.1% (1.3), and 51.9% (1.3), respectively. Greater changes in DMFS were associated with lower education levels, those without dental insurance, a lack of compliance with oral hygiene, a lack of compliance with fluoride use, and a lack of routine dental care prior to RT treatment. Age, sex, and chemotherapy were not associated with changes in caries. Patients with oropharyngeal cancer were associated with a slightly lower rate of increase in DMFS% compared with larynx/hypopharynx [59]. 

A recent review published in June 2022 revealed that there have been over 300 published papers on this topic over the previous 82 years [60]. The reader is referred to this article for more detailed information. In short, the development of caries is related to a high cariogenic diet, poor oral hygiene, lowered oral pH, and resultant microbial shifts. In patients with RC, hyposalivation is a major contributing factor, as is a hypothesized decrease in enamel microhardness [61,62]. Interestingly, the rapid advancement of caries is not associated with pain or sensitivity [63]. Artificial intelligence-based neural network-based analyses suggest that dental status before RT treatment is the greatest predictor of the development of RC after treatment [64]. The development of RC is important because it predisposes patients to the development of dental abscesses and the need for extractions, which ultimately increase the risk of ORN (see the section on ORN below). 

Prevention and treatment include meticulous oral hygiene, topical fluoride applications (5000 ppm pastes, fluoride varnish, and use of fluoride delivery trays), frequent dental cleanings (every 3 months), and use of remineralizing agents such as MI paste.

## 7. Osteoradionecrosis

Osteoradionecrosis (ORN) is a complication that may arise subsequent to RT within the head and neck, particularly affecting osseous structures. The precise mechanisms underlying ORN development are complex and multifactorial, involving radiation-induced vascular damage, fibrosis, a reduction in oxygen supply, and impaired bone cell function [65].

The prevalence of ORN is variable and contingent upon factors such as cumulative radiation dosage, treatment regimen, irradiated site, and individual patient characteristics. Despite advancements in RT modalities such as IMRT, its clinical significance remains persistent, particularly among patients contending with aggressive or recurrent malignancies.

ORN manifests as the emergence of exposed and necrotic bone within the irradiated zone. This clinical state ushers forth a spectrum of weighty consequences, encompassing chronic infectious processes, enduring pain, and the potential for pathologic fractures. While the mandible emerges as the primary locus of ORN, its influence extends to other sites of the maxillofacial skeletal framework. The current incidence rate of ORN is 4–8%, underscoring its enduring pertinence within the contemporaneous medical landscape [66]. The prevalence has significantly decreased due to the evolution of radiation regimens from conventional RT to IMRT. However, proton therapy continues to be significantly associated with a high degree of ORN (10.6%) [67,68,69]. 

The clinical presentation of ORN can vary, ranging from asymptomatic radiographic findings to severe pain, exposed necrotic bone, soft tissue inflammation and swelling, the formation of an abscess or fistula, and even a pathological fracture [70]. The presence of necrotic bone serves as a nidus for bacterial colonization and infection, further exacerbating the condition. The risk factors for developing ORN include higher radiation doses, advanced cancer stages, dental extractions or trauma post-irradiation, and poor oral hygiene. Additionally, coexisting factors such as smoking, diabetes, and compromised vascularity can further increase susceptibility to ORN [70,71]. 

For the evaluation of suspected ORN, plain radiographs, Computed Tomography (CT) scans, and Magnetic Resonance Imaging (MRI) are commonly used modalities to assess bone integrity, soft tissue involvement, and the presence of infection. Radiographic findings in ORN include diverse bone density, irregularities, and coarse trabeculations, which can be easily identified. CT is the optimal modality for highlighting these bone alterations three-dimensionally, aiding in identifying the extent and pervasive pattern of trabecular loss and regions of bony sclerosis [72]. The acute or chronic inflammation associated with ORN can resemble recurrent tumors on Positron Emission Tomography (PET)/CT scans where significant fluorodeoxyglucose (FDG) activity at ORN sites is seen. This may be confused with poor local/regional cancer control or local recurrence. A tissue biopsy may be needed for the further determination of suspicious lesions. MRI findings include modified marrow signals characterized by reduced T1 signals and variable T2 signals. The variability in the T2 signal is thought to stem from changes within the marrow space, varied by acute inflammation (increased signal intensity) and fibrosis (decreased T2 signal intensity) at the time of the MRI investigation [73].

The diagnosis of ORN requires careful clinical evaluation, correlation of imaging findings, and a thorough understanding of the patient’s radiation treatment history. Multiple diagnostic criteria for ORN have been proposed, with the key elements being exposed bone, previous history of RT, absence of recurrent tumor, and a variable minimum period of exposed bone (2–6 months) [74]. The extent of bone involvement is the main factor in differentiating the severity of ORN. The different proposed classifications have different emphases, focusing on one or more elements involved in the final staging, including the location of ORN, the clinical outcome (fractures), accompanying symptoms or features, and management. The comparison of the two commonly used classification systems is summarized in Table 5. 

Shaw et al. recently proposed refining the definition of ORN of the mandible, which built on Notani’s classification by including minor bone spicules (MBS) (defined as exposed bone less than 20 mm^2^). The authors suggested that MBS could be common and be associated with a better clinical prognosis [74]. 

Prevention is paramount, with careful treatment planning and consideration of dental extractions or invasive procedures prior to RT. Maintaining good oral hygiene, regular dental evaluations, and appropriate dental interventions before and after treatment are essential. If the extraction of teeth cannot be salvaged, it is recommended that these extractions be carried out 10 to 21 days before the radiotherapy, ensuring the complete epithelization of the extraction sockets [77]. Furthermore, invasive or prophylactic dental procedures should be performed cautiously. 

Management of ORN requires a multidisciplinary approach involving oncologists, oral and maxillofacial surgeons, and dental professionals. Management is challenging and depends on the stage of the disease. Conservative approaches involve pain management, oral hygiene, and antibiotics in the case of infection. Chlorhexidine 0.12% rinses can be used to reduce bacterial flora in the setting of asymptomatic exposed bones. Penicillin, tetracyclines, or other antibiotics can be used to control active infections secondary to ORN. The course of antibiotics should be comparable to the treatment of osteomyelitis, where the course is longer than that applied for the treatment of odontogenic infection. Treatment alternatives combining pentoxifylline and tocopherol (PENTO) have been used for the treatment of ORN, with a complete remission rate ranging from 16.6% to 100% and an overall stable/improved rate >68% [78]. PENTO were also used for the prophylaxis of ORN, but the evidence for its effectiveness is insufficient [79,80].

For advanced stages, surgical interventions such as sequestrectomy, resection, and reconstructive procedures may be necessary. Hyperbaric oxygen therapy (HBOT) has shown promise in promoting tissue healing and reducing symptom severity, but its effectiveness remains to be further explored [81,82].

## 8. Trismus

Trismus, characterized by restricted mouth opening, can significantly affect the quality of life in head and neck cancer patients, particularly those who have undergone RT as part of their treatment. Head and neck radiation induces fibrosis, leading to tissue stiffness in the area of the temporomandibular joints, masticatory muscles, tongue, and constrictor muscles of the pharynx, resulting in reduced elasticity and subsequent limitations in mouth opening and range of motion. Trismus is defined as a restriction in maximal mouth opening of 35 mm (measured from the incisal edges of maxillary and mandibular central incisors) or less. It is reported that prevalence rates range from 6% to 86%, with higher rates observed in patients receiving higher radiation doses [83,84].

Limitations in maximal mouth opening and a lack of elasticity in the inner aspects of the cheeks and lips may lead to the inability to brush posterior teeth effectively. In extreme cases, patients may not be able to fit a toothbrush between their teeth to brush the lingual/palatal surfaces. They may not have the ability to open wide enough to bite into a sandwich. Severe trismus can lead to significant functional impairment, speech impairment, and esthetics and have a negative impact on the patient’s psychological well-being [85].

The comprehensive management of trismus within the context of head and neck cancer patients necessitates a multidisciplinary strategy that encompasses the expertise of radiation oncologists, oncology nurses, speech therapists, physiotherapists, and dentists. Physical interventions, such as active and passive jaw exercises, manual stretching, and myofascial release techniques, are employed to ameliorate restricted mouth opening and mitigate fibrotic tissue formation [86]. The implementation of pharmacological agents, encompassing muscle relaxants and anti-inflammatory medications, may alleviate muscular discomfort in some cases and decrease muscular rigidity. In patients who experience pain or muscle spasms upon opening, administration of botulinum toxin directly into the affected musculature may present an avenue for mitigating muscle spasms, allowing them to perform jaw exercises more effectively and maintain the maximal opening. In instances of severe trismus, surgical intervention may be contemplated to address fibrotic adhesions and restore better mouth opening.

In a systemic review by Chee et al., trismus interventions involving exercise regimens and jaw rehabilitation devices exhibited comparable efficacy. Notably, emphasizing adherence to any specific intervention protocol emerges as a constructive approach to positively influencing measures of mouth opening in this patient population. Additionally, the integration of low-level laser therapy and low-intensity ultrasound in conjunction with exercise regimens emerges as a potentially beneficial modality for individuals afflicted with trismus [87].

## 9. Surveillance for Recurrent and Second Primary Cancer

Cancer surveillance among survivors of oropharyngeal and head and neck cancer treatment falls into the following categories: (1) index case; (2) local and regional recurrence; (3) metastasis; and (4) second primary. The concept of field cancerization was thought to be responsible for multiple, synchronous primary squamous cell carcinomas mostly attributable to smoking history and helped formulate the surveillance schedules for survivors. According to the American Cancer Society Head and Neck Cancer Survivorship Care Guidelines and other international groups, surveillance should be conducted every 1–3 months for the first year after primary treatment, every 2–6 months in the second year, every 4–8 months in years 3–5, and annually after 5 years [88].

Up to 90% of all recurrences occur within the first 2 years after the completion of cancer therapy [89,90]. The follow-up intervals are developed largely depending on the type of cancer, location within the oral cavity (most suspicious sites being the floor of the mouth > tongue > oral mucosa of the mandible > oral mucosa of the maxilla and cheek), and HPV status. 

The risk for the development of second primary malignant neoplasms (SPMN) is estimated to be 3–7% per year, with a 20-year cumulative rate of 36% [91,92]. Other factors that are considered when determining follow-up intervals include alcohol/tobacco status at diagnosis, with the higher recurrence rates occurring in those with a combination of smoking and alcohol abuse (49%), followed by smoking alone or alcohol alone (13% each), tumor size, and histopathological grade [88,93]. HPV negative patients have a 10-fold increased rate of development of SPMN in the head and neck, lung, bladder, or esophagus after receiving adjuvant radiation with or without chemotherapy [94]. Interestingly, for patients in 3–5-year follow-up, a new disease was found in only 10% of asymptomatic subjects. The asymptomatic detection of new disease was not associated with improved overall survival, meaning that 90% of recurrent or second primary development was symptom-driven [94].

Patients with HPV+ oropharyngeal cancers had a 4.9% risk of developing an SPMN of the upper aerodigestive tract and a 2.2% risk of oropharyngeal SPMN. If adjuvant radiation with or without chemotherapy was delivered, the risk of SPMN decreased by 75%, with no associated changes in overall disease-free survival or overall survival [95]. HPV+ OPSCC patients had a decline in second primary malignancies and a significantly longer median time to development of SPMN [96]. Additionally, while 50% and 75% of local recurrence and distant metastasis occur within the first 200 and 400 days, respectively, the mean time for second primary cancers was 1589 days [97]. These results indicate that less stringent follow-up is needed for HPV+ oropharyngeal cancer survivors and also illustrate the need for a subset-driven surveillance schedule or personalized surveillance. 

## 10. Conclusions

Complications related to the treatment of oropharyngeal cancer are common and have a wide-ranging impact on function, health, and quality of life. Emphasis should be placed on the identification of the anticipated side effects, the application of preventative measures when able, and management following treatment. The side effects involve multiple dental and medical specialties and a multidisciplinary team approach. The findings of this review reveal that side effects and severity vary widely, as do treatment and management guidelines. Evidence used to create treatment guidelines tends to be “best practices,” as more well-designed clinical studies are needed. 

## Figures and Tables

**Table 1 cancers-15-04548-t001:** Stages of oral mucositis.

Stage	Description
Initiation	- DNA damage caused by chemotherapy or radiation therapy - Triggering of oxidative stress responses and innate immune activation - Cell apoptosis and breakdown of epithelial cells
Up-regulation and Activation	- Transcriptional factors activated by reactive oxygen species and innate immune response - Production of proinflammatory cytokines and stress responders - Apoptosis and necrosis in mucosa and submucosa
Signal-Amplification	- Molecules from primary response further amplify inflammation - Feedback mechanisms contribute to tissue damage - Promotes bacterial spread and sustains proinflammatory cytokine production
Ulcerative	- Progressive tissue injury and loss of epithelial continuity - Development of symptomatic, deep ulcers - Ulcers become susceptible to infections - Accompanied by intense inflammation
Healing	- Occurs at the end of treatment - Ulcers heal through epithelial migration, proliferation, and differentiation - Mucosa is restored with altered genetics, potentially less resistance to future treatments

**Table 2 cancers-15-04548-t002:** Comparison of oral mucositis scales.

Grade	WHO Oral Mucositis Scale	CTCAE V5 Oral Mucositis Scale	RTOG Oral Mucositis Scale
1	Soreness/erythema	Asymptomatic or mild symptoms Intervention not indicated	Erythema
2	Soreness/erythema + ulceration + ability to eat solid foods	Moderate pain or ulcer with no interference with oral intake Modified diet indicated	Patchy reaction (<1.5 cm, non-contiguous)
3	Soreness/erythema + ulceration + ability to use a liquid diet only	Severe pain, interfering with oral intake	Confluent mucositis (>1.5 cm, contiguous)
4	Soreness/erythema + ulceration + no possible oral alimentation	Life-threatening consequences Urgent intervention indicated	Ulceration, necrosis, bleeding
5		Death	

**Table 3 cancers-15-04548-t003:** Common antifungal treatment for oral and oropharyngeal candidiasis.

Class/Drugs	Form	Dose
**Topical**		
Nystatin	Suspension (100,000 U/mL)	4–6 mL PO rinse 4–5 min QID for 10 days
	Pastille (200,000 U each)	Dissolve 1 pastille after meals QID for 7–14 days
	Cream (100,000 U/g)	Apply directly to area of infection TID–QID
Amphotericin B	Suspension (100 mg/mL)	100–200 mg PO swish QID for 14 days
	Lozenge (10 mg)	Dissolve after meals TID for 14 days
Ketoconazole	Cream (2%)	BID–TID for 14–28 days
Miconazole	Gel (2%); Cream (2%)	Apply directly to the area of infection TID–QID for 14–21 days
	Mucoadhesive tablets	50 mg QID 14 days
Clotrimazole	Troche (10 mg)	10 mg dissolved PO 5×/day for 14 days
	Cream (1%)	Apply directly to the area of infection BID–TID for 21–28 days
**Systemic**		
Fluconazole	Tablet (100 mg)	Loading dose of 200 mg followed by 100 mg QID for 7–14 days
Itraconazole	Capsule (100 mg)	100 mg QID for 14 days
Ketoconazole	Tablet (200 mg)	200–400 mg QD for 14 days

PO, oral; QID, four times daily; TID, three times daily (adapted from refs [32,33]).

**Table 4 cancers-15-04548-t004:** Guidelines for management of salivary gland hypofunction and/or xerostomia induced by nonsurgical cancer therapies [43].

Recommendation	Evidence Quality/Strength
**Preventive approaches to reduce the risk of salivary gland hypofunction**	
IMRT to spare salivary glands from higher dose radiation	High/Strong
Other radiation modalities that limit cumulative dose to salivary glands	Low/Strong
Acupuncture during RT	Intermediate/Moderate
Systemic administration of bethanechol during RT	Low/Weak
Vitamin E/other antioxidants to reduce the risk of radiation-induced salivary gland hypofunction	Low/weak
Submandibular gland transfer before head and neck cancer treatment	Insufficient
Use of oral pilocarpine, amifostine (with contemporary radiation modalities), or low-level laser therapy	Insufficient
Other interventions *	Insufficient
**Treatments to improve xerostomia/hyposalivation**	
Topical mucosal lubricants or artificial saliva	Intermediate/Strong
Gustatory and masticatory salivary reflex stimulation by sugar-free lozenges, acidic candies, or sugar-free, nonacidic chewing gum	Intermediate/Moderate
Oral pilocarpine, and cevimeline	High/Strong
Acupuncture	Low/weak
Transcutaneous electrostimulation or acupuncture-like transcutaneous electrostimulation	Low/Weak
Extract of ginger and mesenchymal stem cell therapy	Insufficient

* Other agents include: n-acetylcysteine oral rinse, traditional Chinese medicine-based herbal mouthwash, local clonidine, concurrent chemotherapy with nedaplatin, boost RT, hyperfractionated or hypofractionated RT, intra-arterial chemoradiation, minocycline, melatonin, nimotuzumab, zinc sulfate, propolis, viscosity-reducing mouth spray, transcutaneous electrical nerve stimulation (TENS), parotid gland massage, thyme honey, and human epidermal growth factor. Modified from salivary gland hypofunction and/or xerostomia induced by nonsurgical cancer therapies: ISOO/MASCC/ASCO Guideline [43] to answer the question: What are the most effective interventions to prevent, minimize, and manage salivary gland hypofunction and xerostomia in the oncology patient receiving nonsurgical cancer therapy?

**Table 5 cancers-15-04548-t005:** Comparison of ORN classifications.

Stage	Notani Classification [75]	Lyons Classification [76]
1	ORN confined to dentoalveolar bone	<2.5 cm length of bone affected (asymptomatic); medical treatment only
2	ORN is limited to the dentoalveolar bone or mandible above the inferior dental canal, or both.	>2.5 cm length of bone, asymptomatic (including pathological fracture or involvement of inferior dental nerve); medical treatment only unless there is dental sepsis or obviously loose, necrotic bone
3	ORN involving the mandible below the inferior dental canal with pathological fracture, or a skin fistula	>2.5 cm length of bone, symptomatic (but no other features); consider debridement of loose or necrotic bone, and local pedicled flap
4		>2.5 cm length of bone, pathological fracture, involvement of the inferior dental nerve, or cutaneous fistula, or a combination;reconstruction with free flap if patient’s overall condition allows

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
