# Peer review of "Oral Complications from Oropharyngeal Cancer Therapy"

_cancers, 2023, doi:10.3390/cancers15184548_

Round 1

Reviewer 1 Report

The manuscript clearly lists all possible complications related to the therapy. 

It is recommended to correct the summary and write it according to the instructions. Conclusion also should be expanded while references should be written according to the instructions for authors.

Author Response

The manuscript clearly lists all possible complications related to the therapy.

It is recommended to correct the summary and write it according to the instructions. Conclusion also should be expanded while references should be written according to the instructions for authors.

The Summary and Conclusions have been modified to fulfil the instructions for authors.

Reviewer 2 Report

It is an interesting work on the complications from treatment of oropharyngeal tumors.

It is a compilation paper even if quite complete and exhaustive.

The summary tables in which the various classifications are compared are particularly interesting and useful.

In the paragraph relating to osteoradionecrosis, it should be emphasized that it is essential to avoid every trauma in the 18-36 months following the end of treatment. More invasive dental procedures (extractions, implants, etc.) should be performed before starting the radiotherapy and, if necessary, carried out with extreme caution and under prolonged antibiotic prophylaxis.

There are some typing errors:

line 272 pian > pain 

line 312 3 dimentially > 3 dimentionally 

Author Response

In the paragraph relating to osteoradionecrosis, it should be emphasized that it is essential to avoid every trauma in the 18-36 months following the end of treatment. More invasive dental procedures (extractions, implants, etc.) should be performed before starting the radiotherapy and, if necessary, carried out with extreme caution and under prolonged antibiotic prophylaxis.

We appreciate the suggestions from the reviewer. We have added content about the general suggestions about dental extraction before or after the radiation therapy. Regarding the suggested timeline that trauma/extraction should be avoided, we found the evidence is inconsistent. Also, extraction is inevitable if the restorable tooth becomes infected. Thus, in the manuscript, we added: “A comprehensive dental assessment before initiating radiation therapy is pivotal for ORN prevention. If the extraction of teeth cannot be salvaged, it is recommended that these extractions be carried out 10 to 21 days before the radiotherapy, ensuring the complete epithelization of the extraction sockets. Furthermore, invasive or prophylactic dental procedures should be cautiously performed.”

There are some typing errors:

line 272 pian > pain

line 312 3 dimentially > 3 dimentionally

Thank you for pointing this out, the errors have been addressed.

Reviewer 3 Report

This is an interesting manuscript about oral complications of oropharyngeal cancer treatment. The manuscript is well written and is easy to read. I have few suggestions:

Introduction: please provide more information about the different treatment options for oropharyngeal cancer, as it is only mentioned IMRT and proton beam therapy, but no comment on surgery and chemotherapy. Would recommend to expand on this a little bit on the introduction.

Oral mucositis: many treatment alternatives are mentioned, including conventional and non-conventional therapies, and recommendations, such the use of honey. Apart from mentioning those options, I think the article would benefit of providing a standard or first line mucositis treatment.

Page 5 line 183. The ­ acronym ORN is used for the ­­­first time without a definition

Table 4 illustrate preventive strategies to reduce salivary gland hypofunction after RT, and treatment options, therefore I would suggest dividing the table that way. Also, the table needs references.

There are some typos, e.g., Page 280 line 264: patients with oropharyngeal cancer was… should be were, page 280 line 272: it says pian instead of pain, page 10 line 415 says SPM, should be SPMN. Please check the manuscript for other mistakes.

Page 332, the paragraph from lines 300-307 need references.

Page 332 line 315: please provide a definition for FDG

Author Response

This is an interesting manuscript about oral complications of oropharyngeal cancer treatment. The manuscript is well-written and is easy to read. I have few suggestions:

Introduction: please provide more information about the different treatment options for oropharyngeal cancer, as it is only mentioned IMRT and proton beam therapy, but no comment on surgery and chemotherapy. Would recommend to expand on this a little bit on the introduction.

We would like to thank the reviewer for the suggestion, and we have added updated surgical approaches and new treatment modalities and their impact on oral toxicities to the introduction.

Oral mucositis: many treatment alternatives are mentioned, including conventional and non-conventional therapies, and recommendations, such the use of honey. Apart from mentioning those options, I think the article would benefit of providing a standard or first-line mucositis treatment.

We thank the reviewer for the suggestion and agree that current standard therapy is needed. The manuscript has been modified to include current first-line therapies.

Page 5 line 183. The acronym ORN is used for the first time without a definition

Thank you for catching this, it has been defined.

Table 4 illustrate preventive strategies to reduce salivary gland hypofunction after RT, and treatment options, therefore I would suggest dividing the table that way. Also, the table needs references.

We agree with the reviewer’s recommendations and the table has been modified accordingly.

There are some typos, e.g., Page 280 line 264: patients with oropharyngeal cancer was… should be were, page 280 line 272: it says pian instead of pain, page 10 line 415 says SPM, should be SPMN. Please check the manuscript for other mistakes.

Thanks to the reviewer for pointing these out, we have made the changes.

Page 332, the paragraph from lines 300-307 need references.

Thank you for pointing this out, we have added Chronopoulos, A., et al., Osteoradionecrosis of the jaws: definition, epidemiology, staging and clinical and radiological findings. A concise review. Int Dent J, 2018. 68(1): p. 22-30.

Page 332 line 315: please provide a definition for FDG

Thanks to the reviewer for pointing these out, we have made the changes.

Reviewer 4 Report

Thank you for giving me the opportunity to review this narrative review paper on “Oral Complications from Oropharyngeal Cancer Therapy”. It is well-written and informative.

I have some comments and suggestions for the authors to consider:

1.     1.introduction: you mention targeted therapies as one of the reasons for better outcomes in patients with oropharyngeal tumors, but what about immunotherapy directed at the PD1/PD-L1 axis? Please consider to mention immunotherapy in the introduction section.

2.     Line 112 – 121. Please summarize the MASCC/ISOO clinical practice guidelines for mucositis more accurately, since not all guidelines you mention apply to the prevention and treatment of mucositis in HNC patients. Furthermore, the recommendation for glutamine is specifically for oral glutamine and not parenteral administration as this might be associated with higher mortality.

3.     Line 180: CT should read RT.

4.     Radiation caries has a complex etiology. Some studies suggest that RT could directly alter the mechanical properties, micro-morphology, crystal properties, and chemical composition of dental hard tissue. I feel this should be briefly mentioned in section 6.

5.     Dysphagia is a frequent and potentially dangerous complication in patients treated for oropharyngeal cancer. Please pay some attention to this complication.

6.     You may consider to rename section 9 in “Surveillance for Recurrent and Second Primary Cancer”. 

Author Response

  1. introduction: you mention targeted therapies as one of the reasons for better outcomes in patients with oropharyngeal tumors, but what about immunotherapy directed at the PD1/PD-L1 axis? Please consider to mention immunotherapy in the introduction section.

We thank the reviewer for pointing this out, the introduction has been modified to include this.

  1. Line 112 – 121. Please summarize the MASCC/ISOO clinical practice guidelines for mucositis more accurately, since not all guidelines you mention apply to the prevention andtreatment of mucositis in HNC patients. Furthermore, the recommendation for glutamine is specifically for oral glutamine and not parenteral administration as this might be associated with higher mortality.

Thanks to the reviewers, we have added a paragraph summarizing management for oral mucositis.

  1. Line 184: CT should read RT.

 Thanks to the reviewer for pointing these out, we have made the changes.

  1. Radiation caries has a complex etiology. Some studies suggest that RT could directly alter the mechanical properties, micro-morphology, crystal properties, and chemical composition of dental hard tissue. I feel this should be briefly mentioned in section 6.

Thanks to the reviewer for pointing this out, we have added information regarding the effect on the dental hard tissues as requested.  

  1. Dysphagia is a frequent and potentially dangerous complication in patients treated for oropharyngeal cancer. Please pay some attention to this complication.

We appreciate the reviewer's comment and understand that dysphagia is a significant concern in oropharyngeal cancer patients. It often arises due to the proximity of treatment areas to critical swallowing structures in the oropharynx, such as the throat muscles and the epiglottis. Management of dysphagia may involve a range of interventions, including speech therapy, dietary modifications, and surgical interventions which is beyond the scope of this review as the primary focus is oral toxicities associated with cancer treatment.

  1. You may consider to rename section 9 in “Surveillance for Recurrent and Second Primary Cancer”. 

Thank you for the suggestion, the section has been renamed.

Round 2

Reviewer 3 Report

The authors have addressed my comments properly